# Acoustic Sensing Performance Investigation Based on Grooves Etched in the Ring Resonators

**DOI:** 10.3390/mi14030512

**Published:** 2023-02-22

**Authors:** Yuan Han, Yongqiu Zheng, Nan Li, Yifan Luo, Chenyang Xue, Jiandong Bai, Jiamin Chen

**Affiliations:** Key Laboratory of Instrumentation Science and Dynamic Measurement Ministry of Education, North University of China, Taiyuan 030051, China

**Keywords:** acoustic sensor, ring resonator, optical acoustic detection system, frequency bandwidth, high sensitivity

## Abstract

Acoustic detection based on optical technology has moved in the direction of high sensitivity and resolution. In this study, an optical waveguide acoustic sensor based on a ring resonator with the evanescent field is proposed. Grooves are introduced into the ring resonators as a direct sensitive structure to excite the evanescent field. A series of resonators with diverse grooves are fabricated for a comparative analysis of acoustic performance. The acoustic parameters of bandwidth, sensitivity, and signal-to-noise ratio (SNR) vary with different grooves indicated by the Q-factor. The results show that the ring resonators with variable-sized grooves exhibit excellent capability of acoustics detection. A maximum frequency of 160 kHz and a high sensitivity of 60.075 mV/Pa is achieved, with the minimum detectable sound pressure being 131.34 µPa/Hz^1/2^. Furthermore, the resonators with high Q-factors represent a remarkable sound resolution reaching 0.2 Hz. This work is of great significance for optimizing acoustic sensors and broadening the application range.

## 1. Introduction

Acoustic detection [1,2,3] based on optical devices has intensified interest in many application areas, including non-contact measurements [4,5,6], optoacoustic imaging [7,8], industrial manufacture [9,10], and non-destructive testing [11,12]. Conventional acoustic transducers [13] such as piezoelectricity operate over a band of frequencies centered at resonance and their sensitivity relies on the size of the active transducer posing a challenge for miniaturization. However, optical techniques can potentially remove the restrictions of piezoelectric technologies. Optical sensors are immune to the effect of electromagnetic interference making them appropriate for a greater spectrum of environments and have the advantages of being lightweight and possessing mechanical flexibility. Therefore, as an emerging research field, the optical sensor is increasingly considered a prospective alternative to sound signal detection.

Advances in all-optical detection have been recently revealed, in particular, in optical resonators. The resonator focuses the detection beam within a small volume, enhancing the interaction between the beam and the acoustic wave, thus increasing the sensitivity to acoustic fluctuations. The classical optical resonator geometries include Fabry–Pérot interferometers [14,15,16], fiber Bragg gratings [17], and ring resonators [18,19,20]. The XARION Laser Acoustics Company [21] has realized acoustic detection by means of an interferometric cavity formed by two parallel semi-reflective mirrors. The novelty principle is based on the modulation of light caused by the small changes in the refractive index of air induced by sound signals leading to an outstanding acoustic detection bandwidth. Moreover, a highly sensitive compact hydrophone based on the fiber Bragg grating has been developed [17]. The grating displays a sharp resonance with a central wavelength that is sensitive to pressure. This resonance is monitored by a continuous wave laser to measure the pressure changes within the grating caused by the acoustic wave. Unfortunately, they fail to build on Si-based platforms using photonic integration.

The ring resonator [22,23,24,25] plays an important role in photonics; it is comprised of a ring structure and a coupling to the outside world. It is quite prolific in practical applications attributed to a single structure and the availability of complementary metal-oxide-semiconductor (CMOS) fabrication technology. Song et al. designed and fabricated on-chip Ge_11.5_As_24_Se_64.5_ microcavity resonators [26]. The polymer material, Ge_11.5_As_24_Se_64.5_, with low Young’s moduli, will distort when the ultrasound waves impinge on the surface of an on-chip microcavity resonator. The strain induced by the acoustic vibration can change the refractive index (RI) of the mode through the elastic-optic effect. However, the performance demonstrated in the experiment is only at 40 kHz and the sensitivity is expressed by the signal-to-noise ratio (SNR) of just 2.86 dB/Pa. Moreover, in 2020, the etched SOI microring resonator for ultrasound measurement was designed by the University of Sydney [27], depicting a selectively etched microring resonator in an add-drop structure. It provides a linear and distortion-free sensing response to the ultrasound signal. Nonetheless, the resolution of grooved acoustic sensing in the uncoupled area is less than ideal, only 30 Hz. In consequence, ring resonators with superior performance need to be further developed.

The purpose of this work is to investigate the acoustic sensing performance based on ring resonators using silicon-based optical waveguides with variable-sized grooves. The grooves can be utilized to excite the evanescent field, therefore allowing it to interact with the atmosphere. The sound wave, propagating in the air, will influence the evanescent wave of the resonator on account of the air refractive index variation, causing the change of the effective refractive index in the waveguide which will be detected based on the phase modulation spectroscopy technology to indicate the acoustic sensing. The phase modulation spectroscopy technology, i.e., phase modulation and lock-in amplification technique, is employed for the sound signal detection. The synchronization signal of the modulation signal is used as the reference signal, then the lock-in amplifier amplifies and extracts the measured sound signal. We demonstrated the physical properties of ring resonators using waveguides with different grooves, followed by a comparative analysis of the variations in the acoustic capability of the frequency range, sensitivity, and SNR.

## 2. Sensor Design and Operation Principle

Light is guided and propagated through the waveguide using total internal reflection and the guided modes will leak from the core of the waveguide into the cladding material as evanescent waves, which will become evanescent fields. If the waveguide is contained within a specific device structure, such as a ring resonator, or Mach–Zehnder interferometer, the refractive index change of the mode in the waveguide can be precisely detected. In this way, it is translated into a wavelength shift of the resonance, or into a change in the intensity of the light transmitted at a given optical wavelength. The sound waves are longitudinal as they travel through the air. In the case of ring resonators, when partial cladding material is air, sound waves change the density of the air resulting in a change in the refractive index of the cladding’s air domain. Ultimately, it translates into the variation of the effective refractive index in the waveguide indicated by the resonance wavelength shift being detected.

The preparation process used inductively coupled plasma (ICP), plasma-enhanced chemical vapor deposition (PECVD), and lithography techniques, which are well-developed to ensure universal applicability. Moreover, the local etching is innovatively added at the end, which can guarantee the acoustic wave adequately interact with the evanescent wave after the evanescent field is excited. The next step is optical fiber coupled with a waveguide via the aligned coupling platform and fixed by optical adhesive. Finally, we put it on the stove for more than 8 h to solidify. The resonator structure is simple, easy to process and design, being applicable to sensing the sound. Figure 1 shows the cross-section of the groove region, the 3D structure diagram, and the finished product.

In this structure of optical waveguide, total internal reflection occurs at the interface between the core and cladding, making light propagation limited in the core layer due to the maximum refractive index of the waveguide core. However, when the groove is introduced, the evanescent field represents an exponential decay in the domain of the groove. The cladding material is silica, and germanium ion is incorporated into the core to control the refractive index difference. Because the difference in refractive index between the cladding and core is 0.75%, the waveguide resonator has a weak refractive index, and the groove increases the area of interaction between light and the surrounding medium. Therefore, the evanescent field extends further into the surrounding environment due to the low mode limitation inside the core material. Both of these effects improve the sensitivity of resonator sensing. Moreover, because of the waveguide with a weak refractive index, we choose a partial groove to ensure that there is not too much light loss to generate resonance, which equips it with sensitive sensing at the same time. The ring circumference is designed to be 9.5 cm, and the coupling length is 100 µm. The minimum etched region is 130 × 130 µm^2^, and the maximum etched region is 160 × 160 µm^2^. The etched region is numbered in increments of 10 µm from small to large. The different etched regions in the two extreme limits are shown in Figure 2.

According to the process characteristics, the parameters of the resonators are tested and compared, and the results are shown in Table 1. On account of the resonators having high Q-factors, the laser used in the experiment is a narrow linewidth laser (E15, central wavelength 1550 nm, NKT Photonics, Birkerød, Denmark). Under the scanning of the external signal source, the frequency of the output light wave changes periodically with the external source. The full width at half maximum (*FWHM*) of the resonance and extinction ratio (*ER*) can be obtained by calculating the corresponding voltage difference of the scanning signal.
(1)$$FWHM=100\times \Delta V\times K_{PZT}$$
(2)$$Q=\frac{\nu_{0}}{FWHM}$$
where *K_PZT_* is the tuning parameter of the laser; the one used in the experiment is 15 MHz/V. 100 is the magnification of the high voltage amplifier (HVA). $\nu_{0}$ is the central frequency of laser.
(3)$$ER=\frac{T_{\max}-T_{\min}}{T_{\max}}$$
where *T_max_* is the maximum value of the resonant spectral line, *T_min_* is the minimum value of the resonant spectral line, and *T_max_* − *T_min_* is the amplitude range of the resonant spectrum. The resonant depth *ER* is related to the loss and replenishment of optical energy in the cavity. When the loss and replenishment of optical energy in the cavity reach dynamic balance, the output light intensity of the cavity is 0, and the resonant depth reaches the highest, *ER* = 1. The measured property spectrum of the resonator is shown in Figure 3.

## 3. Acoustic Measurement and Discussion

As indicated by the above test data in Table 1, the Q-factors of the processed resonators are all up to 10^6^, while the ring circumference and coupling length remain unchanged. With the increase of the groove size, the light leakage will also augment, and the Q-factor and ER show a downward trend. Although the groove will reduce the Q-factor, meanwhile, it will increase the sensing area. Considering the subsequent sound sensing, it is necessary to maintain a balance between the nature of the resonator itself and the acoustic sensing performance. Furthermore, it constructed an acoustic signal detection system based on phase modulation spectrum technology. The test system is composed of the laser as mentioned before: a phase modulator (PM, Eospace, Redmond, WA, USA), a photodetector (PD, New Focus 2053, Shanghai, China), a lock-in amplifier (LIA, SR844, Stanford Research Systems, Sunnyvale, CA, USA), a signal generator (SG, AFG31000, Tektronix, Beaverton, WA, USA), a high voltage amplifier (HVA), a frequency spectrum analyzer (SA, N9030A, Keysight, Santa Rosa, CA, USA), an oscilloscope (OSC, MDO4104C, Tektronix, Beaverton, WA, USA), and proportional integral (PI) module, as shown in Figure 4.

The phase modulation spectroscopy technology is the crucial technique employed for the sound signal detection. The signal detection module based on this technique is LIA which can realize the demodulation of the synchronous signal and improve SNR in the acoustic measurement system. The lock-in amplifier module consists of the signal channel, reference channel, phase sensitive detector (PSD), low pass filter (LPF), and output gain. In the LIA system, the signal to be measured is derived from the PD containing the carrier modulation information, which is amplified, filtered, then transmitted to the PSD. The reference signal is the carrier signal driving the phase modulator. It is synchronously triggered and phase-shifted, made itself in the same frequency and phase as the signal to be measured, and finally, transmitted to the PSD. The two signals are mixed in the PSD and output through the LPF and output gain. The LIA can realize a weak signal extraction in the signal to be detected.

The laser is controlled by the scanning signal derived from the PI module through the HVA. The beam from the laser passes through the PM and enters the ring resonator. Then, the beam of the ring resonator goes into the PD to complete the photoelectric conversion. The LIA receives the signals from the PD and SG for demodulation. When the lock switch of the PI module is in the disabled state, the output frequency of the laser changes linearly under the scanning of the triangular wave, and the resonator outputs the resonant spectrum line with periodic changes. Synchronous demodulation is realized through the LIA. When the lock switch of the PI module is off, the laser makes the resonance spectrum line slowly enter the resonance valley under the action of the scanning signal, then the digital proportional integration control module (PI) starts after the circuit captures the resonance valley signal. The resonance curve, demodulation curve, and resonance valley variation are synchronized, and the PI module control laser only works in the linear region of the demodulation curve; as a result, the laser frequency fails to mistakenly lock outside the resonance valley. The demodulation curve as an error signal to feedback controls the laser frequency, and the feedback control circuit of the PI module changes the laser-scanning voltage. In consequence, the error signal is reduced until zero; at this time, the laser frequency is statically locked at the resonant point, as shown in Figure 4b. After frequency locking, the response of the sensor to the acoustic signal is reflected as the deviation of the relative zero of the demodulation curve. The source of the acoustic system is composed of a loudspeaker, power amplifier (PA), and SG. When an acoustic signal adds to the system, the demodulation curve shows an apparent amplitude change, as shown in Figure 4c, and is connected to the SA to obtain a direct image of the frequency after the FFT transfer.

The frequency response range of acoustic signals is divided into 0.05–20 kHz, generated by the loudspeaker, and ultrasonic frequencies above 20 kHz are tested by switching to piezoelectric ceramics as a result of the acoustic source frequency limitation. In the frequency response experiment, the output amplitude of the SG is kept at 10 V to maintain the stability of the sound pressure level. Song et al. calibrated the performance of the polymer ring resonator [26] at 40 kHz, and the SOI microring resonator produced by Yang et al.’s [27] response frequency points are tested at 40 kHz, 58 kHz, 200 kHz, and 300 kHz to indicate the frequency range. On this basis, we lessen the frequency interval. The interval is narrowed to 10 Hz in the range of 50 Hz to 100 Hz. The interval is 100 Hz between 100 Hz–1 kHz, and the interval is 1 kHz in the range of 1 kHz to 20 kHz. The ultrasonic frequency range is increased appropriately to test the frequency range in which the proposed sensor can react. The frequency response range of the ring resonators is collected employing a spectrum analyzer: acoustic signal response data are collected at a certain frequency point until the amplitude of the sound signal fails to be distinguishable from background noise. The test results are shown in Figure 5. Note that every sensor is equipped with a flatness of ±2 dB, and the widest frequency range achieved is 160 kHz. The resonator capability is better when the Q-factor is larger, and a wider frequency response is accomplished. Nevertheless, certain high-frequency responses can still be observed with a stable amplitude during the ultrasonic frequency measurement but they are incapable of being differentiated from the background noise signal. Therefore, the absence of a large frequency can be considered as loud system noise, an intensive electrical connection, and a too-small SNR, which cannot be tested. Subsequent noise reduction processing will be carried out through the DSP module.

Sensitivity is one of the main exhibitions of acoustic performance, which is expressed as the fitting slope corresponding to the ratio of the output voltage value of the sensor and the sound pressure measured by the standard sound level meter (AWA5661), expressed in units (mV/Pa). The SG generates a 1 kHz sinusoidal signal and remains consistent, and after amplification by the power amplifier, the signal is connected to the sound system. By controlling the amplitude of SG, the acoustic source outputs signals with different sound pressure levels. In the experiment, the SG amplitude increases successively from 1–10 V with a step of 1 V. The voltage output of the sensor on the oscilloscope and the sound pressure value of the standard sound level meter are recorded, respectively. As shown in Figure 6, with the increase of the driving voltage of the sound source, the sound pressure level is gradually enhanced, resulting in the noted variation of the refractive index, and finally, the detection signal rises. It is pronounced that the linearity is great due to all R^2^ being above 0.97. Moreover, the output voltage of the sensor will show a drastically growing trend if the Q-factor is higher. As a result, the sensor with a Q-factor of 3.44 × 10^6^ represents a high sensitivity of 60.075 mV/Pa. The sensitivity of the electric acoustic sensor is high, but the frequency response is limited by the resonant characteristics of the diaphragm, and the optical fiber acoustic sensor based on the diaphragm is similar. At present, although the sensing technology of ring resonators based on detecting the change of the air refractive index caused by sound pressure is not as sensitive as the acoustic sensor relying on a sound-sensitive diaphragm, it is predominant in a good linear response frequency, without frequency dependence and the influence of the mechanical resonance frequency.

Another performance index corresponding to sensitivity is the minimum detectable sound pressure (*MDP*). In the experiment, after giving the speaker a sine sound signal of 1 kHz, the frequency spectrum analyzer is tuned to the corresponding range to collect data. *P_in_* is the real-time sound pressure level obtained by a standard sound level meter, and RBW is the resolution bandwidth of SA. Both *RBW* and *SNR* can be accessed through the relevant parameters on the spectrum analyzer. The noise floor is the average of the environment noise without the sound signal. In the experiment, the noise floor is obtained by estimating the average value of the frequency range other than the acoustic response frequency. *SNR* is the difference between the peak and noise floor. Yang et al. proposed the SOI microring resonator [27] which has an *SNR* of over 50 dB at the measured frequency of 58 kHz. In the paper, we measured frequency at 1 kHz with SNR over 54 dB and calculated *MDP* further. The *MDP* measurement is limited by the sensitivity of the system as a result of the sophisticated signal demodulation system required by optical detectors. On the other hand, the *MDP* is subject to the noise of the physical measurement environment, which can be mitigated if the measurement is carried out in an anechoic chamber to further reduce the minimum detectable sound pressure. It can be reflected from Figure 7 that the ability to pick up weak sound signals with sensitivity increases. The equation for calculating MDP is defined as follows:(4)$$MDP=\frac{P_{in}}{10^{SNR/20}\times \sqrt{RBW}}$$

Moreover, in order to study the sensing resolution, acoustic signals with similar frequencies are applied to the sensor. The SOI microring resonator [27] is able to distinguish 30 Hz. In this paper, we reduce the frequency difference gradually. In the results, the sensors can accurately distinguish 0.2 Hz. Although the performance of the instrumentation used in the experiment will limit the resolution to some extent, the resolution of the sensor itself is still worth exploring in the range that the equipment can achieve. If the resolution of the sensor itself is not accurate enough, the frequency of the sound signal it captures is also less accurate in recognizing small differences in frequency, and the restored sound will be significantly distorted. Figure 8 proves the shift of the frequency spectrum when an acoustic frequency change of 0.2 Hz is brought in, that all resonators can achieve. It is indicated that the proposed sensor framework can distinguish minute frequency differences, making it suitable for spectrally chaotic environments with accurate frequency measurements of acoustic waves required.

From the previous Table 1, it can be seen that as the grooved area increases, the Q-factor represents a downward trend with the proportion of evanescent field increases. The Q-factor is one of the most important performance parameters for ring resonators which determine the acoustic performance. As a result, the acoustic parameters of the frequency range, the sensitivity, and the SNR are proportional to the Q-factor of the ring resonators, yet the minimum detectable sound pressure (MDP) is exactly the contrary, which is consistent with the sensing principles and is clearly reflected in Table 2.

## 4. Conclusions

In summary, based on the acoustic sensing effect of optical waveguide resonators with the evanescent field, we compare the parameters of ring resonators and the acoustic performance. It can be observed that larger grooves will degenerate the resonators’ properties, particularly for the Q-factor. The sensors’ responses to acoustic input are interrelated with the Q-factor; i.e., the frequency band, the sensitivity, and SNR have a positive correlation with the Q-factor, while MDP exhibits a negative correlation. The highest frequency signal and sensitivity can reach 160 kHz and 60.075 mV/Pa, respectively. The minimum detectable sound pressure is contrary to the sensitivity change and reaches 798.72 µPa/Hz^1/2^ at the maximum groove as the SNR exceeds 54 dB. This represents a major advance in acoustic sensors based on optical waveguide resonators with the evanescent field. The goals of our future work are to design different shapes of the ring structure to increase the ring length without adding extra area to increase the Q-factor, thus achieving wider bandwidths, higher sensitivities, and larger SNR with the advantages and flexibility of the proposed design, making it suitable for more applications.

## Figures and Tables

**Figure 1 micromachines-14-00512-f001:**
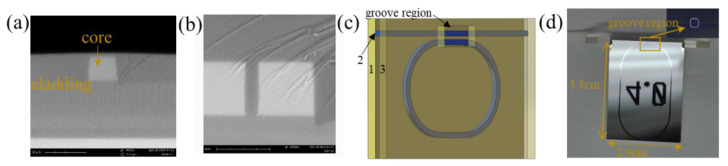
(**a**) SEM image of the cross-section of groove region. (**b**) SEM image of the cross-section of a region without groove. (**c**) Schematic illustration of ring resonator structure: layer 1 and layer 3 are silica cladding layers, and layer 2 is a germanium-doped silica core layer. (**d**) Digital photo of the ring resonator.

**Figure 2 micromachines-14-00512-f002:**

(**a**) The resonator with groove of 130 × 130 µm^2^; (**b**) the resonator with groove of 160 × 160 µm^2^.

**Figure 3 micromachines-14-00512-f003:**
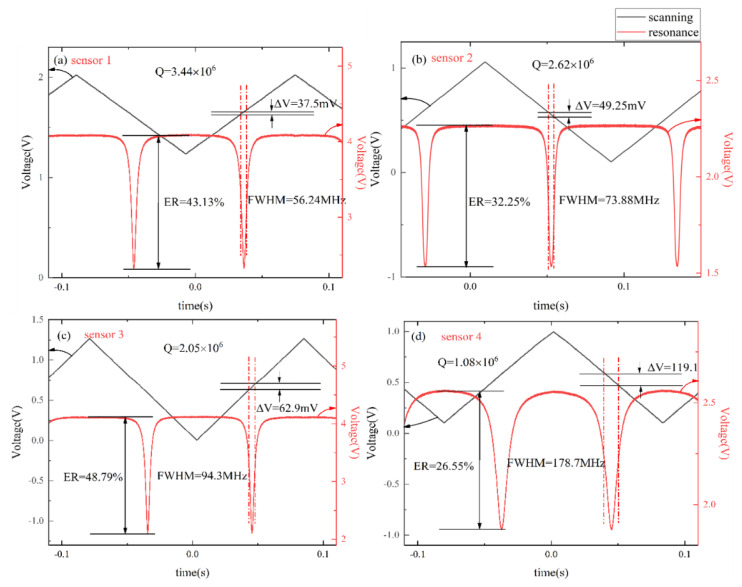
The FWHM and ER property of ring resonators: (**a**) Q = 3.44 × 10^6^ with the groove of 130 × 130 µm^2^; (**b**) Q = 2.62 × 10^6^ with the groove of 140 × 140 µm^2^; (**c**) Q = 2.05 × 10^6^ with the groove of 150 × 150 µm^2^; (**d**) Q = 1.08 × 10^6^ with the groove of 160 × 160 µm^2^.

**Figure 4 micromachines-14-00512-f004:**
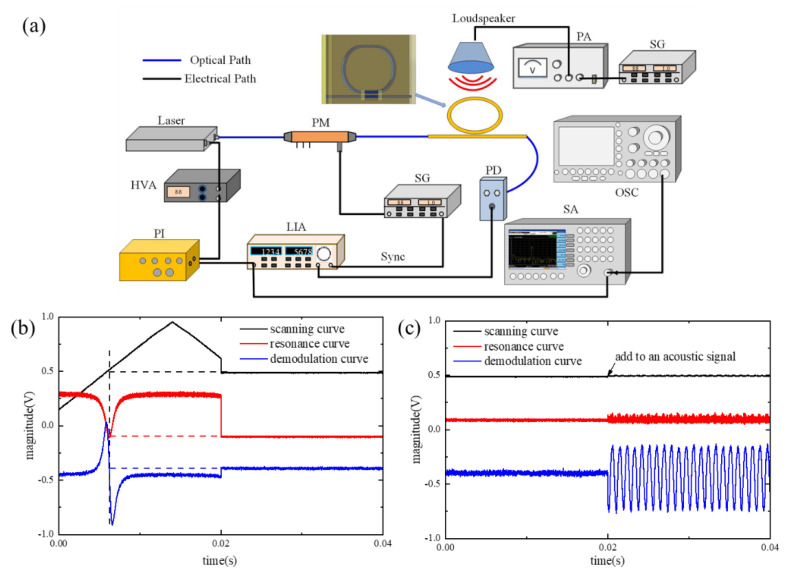
(**a**) The experimental test system of the ring resonators. PM, phase modulator; SG, signal generator; PD, photodetector; LIA, lock-in amplifier; PI, lock-frequency controller; HVA, high voltage amplifier; OSC, oscilloscope; SA, frequency spectrum analyzer; PA, power amplifier. Here, blue lines are the optical path, and black lines are the electrical path. (**b**) Optical spectral lines and frequency-locking curves of the ring resonator of sensor 2. (**c**) The response of demodulation signal of sensor 2 when an acoustic signal is added into.

**Figure 5 micromachines-14-00512-f005:**
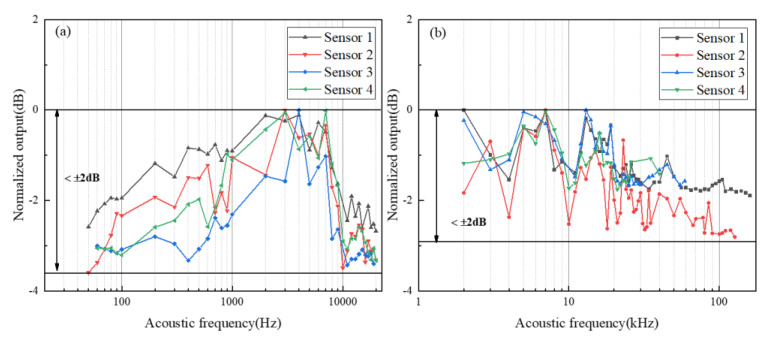
The frequency response and flatness of proposed sensors: (**a**) the frequency response with the loudspeaker; (**b**) the frequency response with the piezoelectric ceramics.

**Figure 6 micromachines-14-00512-f006:**
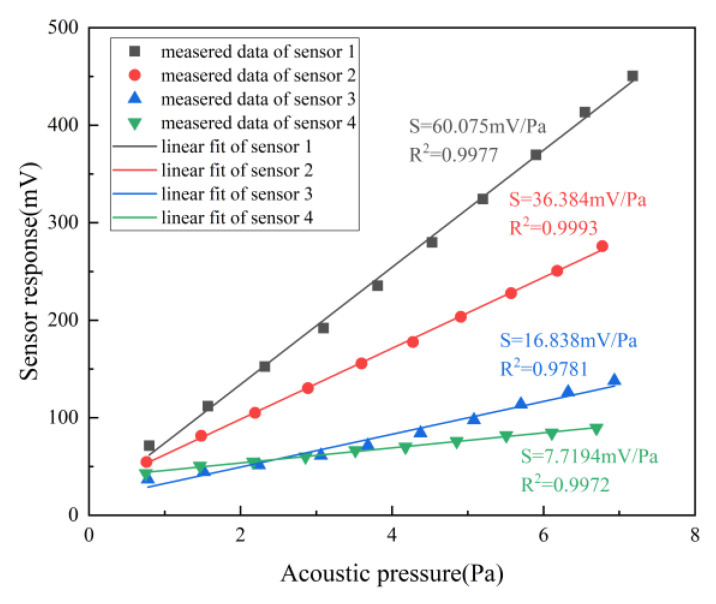
Comparison of the sensitivity of proposed acoustic sensors.

**Figure 7 micromachines-14-00512-f007:**
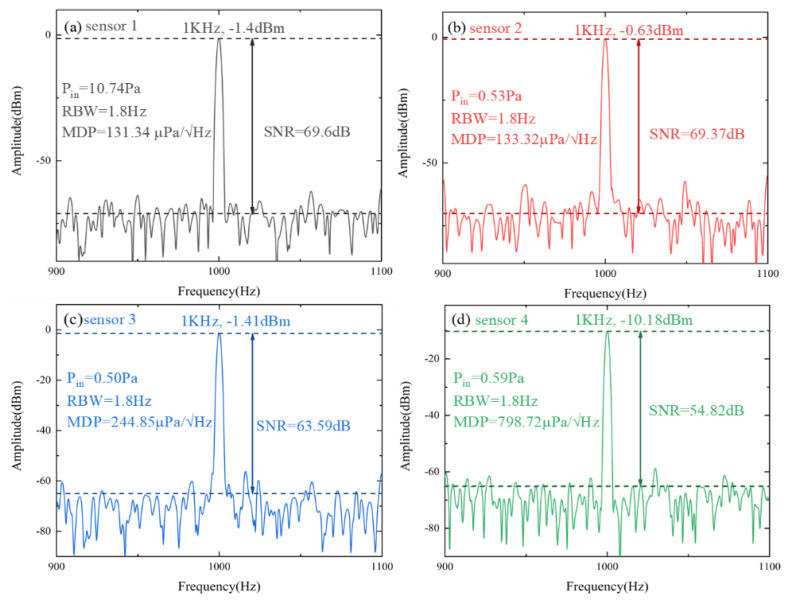
The MDP and SNR characteristics of the sensors at 1 kHz: (**a**) Q = 3.44 × 10^6^ with the groove of 130 × 130 µm^2^; (**b**) Q = 2.62 × 10^6^ with the groove of 140 × 140 µm^2^; (**c**) Q = 2.05 × 10^6^ with the groove of 150 × 150 µm^2^; (**d**) Q = 1.08 × 10^6^ with the groove of 160 × 160 µm^2^.

**Figure 8 micromachines-14-00512-f008:**
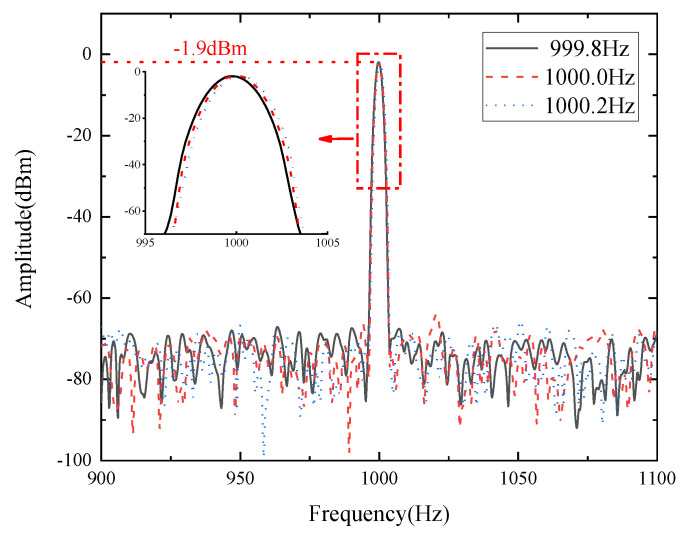
Sensing resolution of the FFT-processed frequency spectrum at the center frequency of 1000.0 Hz with a shift of 0.2 Hz.

**Table 1 micromachines-14-00512-t001:** The parameters of the manufactured ring resonators.

Sensor	Etched Region (µm^2^)	Q	ER
1	130 × 130	3.44 × 10^6^	43.13%
2	140 × 140	2.62 × 10^6^	32.25%
3	150 × 150	2.05 × 10^6^	48.79%
4	160 × 160	1.08 × 10^6^	26.55%

**Table 2 micromachines-14-00512-t002:** The acoustic performance of the manufactured ring resonators.

Sensor	Q	Frequency Range (Hz)	Sensitivity (mV/Pa)	MDP (µPa/Hz^1/2^)	SNR
1	3.44 × 10^6^	50–160,000	60.075	131.34	69.60
2	2.62 × 10^6^	50–127,000	34.384	133.32	69.37
3	2.05 × 10^6^	60–59,000	16.838	244.85	63.59
4	1.08 × 10^6^	60–40,000	7.7194	798.72	54.82

## Data Availability

The data presented in this study are available upon request from the corresponding author.

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
