# Peer review of "Acoustic Sensing Performance Investigation Based on Grooves Etched in the Ring Resonators"

_micromachines, 2023, doi:10.3390/mi14030512_

Round 1
Reviewer 1 Report
The paper addresses a very interesting topic, like the acoustic detection based on optical technology. Please, find my suggestions:
- Please, introduce a brief description of the phase modulation spectroscopy principles in the Introduction
- Please, clarify the acronym when they are firstly used: CMOS (at row 57), RI (at row 57), SNR (at row 59), ICP, PECVD (at row 88)
- Please, clarify the expression "attaching to the atmosphere" at row 68
- I suggest to add the description of each layer in Figure 1(a) and to provide a picture of region with groove and a region without groove
- Please, provide the overlapping factors at different groove between light and the surrounding medium at row 107
- Please, provide a picture of the different etched regions in the two extreme limits at row 114
- Please, provide formulas describing how FWHM and ER were calculated.
- Please, replace "106" with "106" at row 131
- Please, specify in the caption which sensor was tested in Figure 3 (b) and (c)
- Please, explain why the offset signals of the resonance curves in Figure 3 (b) after 0.02 s and (c) are different.
- Please, use as a lower limit of -4dB on the y axis of Figure 4 to better distinguish curves
- Please, check the caption of Figure 4: 4 curves ar shown in the picture but only 2 were described
- Please, provide and comment the different offsets of the sensitivity curves in Figure 5
- Please, replace "Pin" with "Pin" at row 202
- Please, specify how you define the baseline to calculate the SNR in Figure 6
- Please, provide the position of the measured peak in Figure 7
Reviewer 2 Report
In this manuscript, the authors anayzed a ring-shape device for aoucsitc sensing. Conceptual illustration, experimental results were presented. It was claimed that the device has promising acoustic detection capability. This is a well written manuscript, it should be interesting to the readers in this field.
Comments:
1. Performance comparision should be carried out with the results in previous ring-shape sensors. What is the advantage/difference of the proposed devices?
2. What is the size of the device? marker is suggested to be added in Fig. 1
3. From the abstract, it seems that high Q is recommended. However, a high Q device yields smaller deltaV as suggested in Fig. 2.
4.Where is the source of the acoustic in Fig. 3?
Round 2
Reviewer 1 Report
I thank the authors for the careful corrections and I only suggest to replace "Figure 4" at row 278 with "Figure 5".
Reviewer 2 Report
The authors have addressed all the issues